# Monitoring Perfusion Index in Patients Presenting to the Emergency Department Due to Drug Use

**DOI:** 10.3390/jpm13020372

**Published:** 2023-02-19

**Authors:** Gabriela Raluca Grigorasi, Mihaela Corlade-Andrei, Irina Ciumanghel, Ivona Sova, Teofil Blaga, Claudiu Carp, Diana Cimpoesu

**Affiliations:** 1Faculty of Medicine, University of Medicine and Pharmacy “Gr. T. Popa”, 700115 Iasi, Romania; 2Emergency University Department “St. Spiridon” Hospital, 700111 Iasi, Romania

**Keywords:** emergency department, perfusion index, peripheral perfusion, pulse oximeter, organ failure

## Abstract

(1) Background: The perfusion index (PI) represents the ratio between pulsatile blood flow and non-pulsatile blood flow in the peripheral tissue. (2) We aimed to investigate the blood pressure perfusion of tissues and organs in ethnobotanical, synthetic cannabinoid and cannabis derivative consumers through the value of perfusion index. (3) Results: The patients enrolled were divided into two groups: group A, which included all patients who presented in the ED within the first three hours after consumption, and group B, which included those patients who presented more than three hours and up to 12 h after drug consumption. The average values of the PI in the case of group A/group B were 1.51 ± 1.07/4.55 ± 3.66. Statistically significant correlations in both groups were recorded between the drug intake ED admission, respiratory rate, peripheral blood oxygen saturation and tissue perfusion index (*p* < 0.001). The average value of the PI measured in group A was significantly lower compared to that measured in patients in group B. Therefore, we concluded that the perfusion of peripheral organs and tissues was lower in the first 3 h after drug administration. (4) Conclusions: PI plays an important role in the early detection of impaired organ perfusion and in monitoring tissue hypoxia. A decreased PI value may be an early indication of decreased perfusion organ damage.

## 1. Introduction

Drug consumption, especially among teenagers, is recognized as a major public health problem, with evidence that it has significant neurological and psychosocial health consequences. The past decade has seen a near tenfold increase in the THC content of marijuana as well as the increased availability of highly potent synthetic cannabinoids (SCs) for recreational use. This increase in consumption, especially by young people, has made SCs a health problem and a threat to society. Most synthetic cannabinoids used for recreational use are full agonists of CB1R (THC is a partial agonist) with up to several hundred-fold higher potency and efficacy than THC, causing more dangerous adverse effects [1]. SCs manifests their effects by acting on cannabinoid (CB) type 1 (CB1) and CB type 2 (CB2) receptors, [2] through which SCs block the N- and P/Q-type calcium channels, decreasing the membrane potential with sodium–potassium pump inhibition through the interaction with the cell membrane of voltage-dependent ion channels [3]. SCs also exhibit a dose-dependent biphasic effect on the autonomic nervous system. At low and moderate doses, they increase sympathetic activity, reduce parasympathetic activity, cause tachycardia and produce an increase in cardiac output. Conversely, at high doses, they inhibit sympathetic activity and increase parasympathetic activity, causing bradycardia and hypotension [4].

The principle of pulse oximetry is the difference in the absorbance of light with different wavelengths (660 and 940 nm) by oxygenated hemoglobin. Other tissues, such as connective tissue, bone and venous blood, also absorb light and, thus, affect the resulting signal. However, whereas the arterial component of the signal is pulsatile, the absorption of light by other tissues is fairly constant. Therefore, to have a proper estimate of the arterial oxygen saturation of the hemoglobin, the pulse oximetry has to distinguish the pulsatile component from the non-pulsatile component, where the pulsatile component is used subsequently to calculate the arterial oxygen saturation.

Emergency medicine doctors must have a good knowledge of the effects generated by ethnobotanical, cannabis and any other type of drug consumption because patients generally present in emergency departments in the acute phase of symptoms after drug consumption.

This study aimed to investigate the blood pressure perfusion of tissues and organs in ethnobotanical, synthetic cannabinoid and cannabis derivative consumers and to monitor tissue and organ perfusion through the medium of the perfusion index and clinical status of patients admitted to the Emergency Department of St. Spiridon Iasi.

## 2. Material and Method

### 2.1. Study Design

This research was performed as a single-center retrospective cross-sectional study.

### 2.2. Setting

The study was performed in the north-eastern region of Romania, in Iasi County, and was conducted for a 24 month period between 1 January 2017 and 31 December 2019. This retrospective study is the first to be realized in Romania (six countries, over 4.75 million inhabitants) over a prolonged period (24 months) and is one of only a few attempts to analyze the role of the tissue perfusion index (PI) in detecting severe complications in drug users.

### 2.3. Participants

In this study, we included all patients admitted to the Emergency Department “St. Spiridon” Iasi for voluntary/accidental exposure to cannabis, synthetic cannabinoids and new psychoactive (ethnobotanical) substances. We used the data recorded in the clinical observation files. The inclusion criteria were: aged over 18 years, anamnestic data, clinical examination and specific symptomatology to exposure to the above-mentioned drugs and patients with toxicological and laboratory determinations suggestive for the diagnosis of cannabis, synthetic cannabinoids and new psychoactive intoxication (ethnobotanical).

Patients with incomplete data recorded in the clinical observation files were excluded from the study. Additionally, those patients whose index perfusion tissue could not be determined upon admission to the emergency department or who were initially consulted in another health unit and later directed to St. Spiridon Hospital Iasi were excluded from the study.

Other exclusion criteria were: history of vascular diseases (diabetes, hypertension), patients with circulatory failure associated with hypovolemia and low cardiac output.

A urine drug screening test using an immunoassay (Triage Meter Pro) was used to screen for amphetamines, barbiturates, benzodiazepines, cocaine, methadone, methamphetamines (including MDMA), opiates, phencyclidine (PCP), tricyclic antidepressants and tetrahydrocannabinol (cannabis).

### 2.4. Variables

Only patients exposed to the drugs mentioned above were included in the study. Patients who consumed cannabis, synthetic cannabinoids or new psychoactive substances in association with another type of drug were also not eligible for this study.

### 2.5. Data Sources/Measurement

Immediately after admission to the emergency department, all patients benefited from a complete medical examination and the measurement of the parameters included in the study. It is known that most people seek specialized help immediately after exposure due to the intensity of the symptoms. Systolic blood pressure (SBP), diastolic blood pressure (DBP), heart rate (beats/min), Glasgow coma score (GCS) and tissue perfusion index (PI) were collected. After the patients were admitted to the ED, PI was measured by using a noninvasive probe (compact pulse oximeter BM1000E1). The probe was placed on the second, third or fourth digit of the right hand, and the value was recorded after getting a value fixed on the monitor or after waiting for 10 s. The nails were without any varnish, and the fingers were without any temporary or permanent tattoo. It is also known that pulse oximeter measurements can be negatively affected in the presence of high ambient light and a low body temperature. It is a non-invasive technique that allows the continuous measurement of peripheral perfusion using the pulse oximetry, with the pulse oximeter being placed at the level of the tip of a finger (which is generally well perfused with oxygenated blood) or at the level of the earlobe [5]. This index shows the blood level of oxyhemoglobin and is an important parameter for the circulatory and respiratory system. Theoretically, the values of this type of index are between 0.02% and 20%. When the signal is weak, for example, during vasoconstriction, the pulse oximetry signal requires amplification up to ×109. The amplification necessary during a low signal (vasoconstriction, hypovolemia) could limit its clinical application in critically ill patients [6].

### 2.6. Bias, Quantitative Variables

Symptoms in acute poisoning with new psychoactive substances are varied. The onset of symptoms also depends on the consumption type. In the case of inhalation or smoking, they will start immediately after consumption, the maximum intensity will be at 20–30 min and will persist for 3–4 h. In the case of ingestion, they start at 30–60 min to 6 h, and in the case of injection, they can begin almost immediately in the minutes following administration. Considering this, the patients were divided into two groups, A and B, depending on the amount of time that had elapsed between drug consumption and presentation to the emergency department, and statistical data were compared. Group A: The group of patients who presented themselves in the ED in the first 3 h after drug intake. Group B: The group of patients who presented more than 3 h after using the drug. The groups were compared in terms of symptoms, findings, duration of symptoms, time to discharge and hospitalization rates.

### 2.7. Study Size

Being one of the largest centers in the county, we can state that all eligible patients exposed to cannabis, new psychoactive substances and synthetic cannabinoids were included in this study. One eligibility criterion was represented by the first medical consultation that had to take place within the ED “St. Spiridon”.

### 2.8. Statistical Methods

The collected data were statistically analyzed with the support of Microsoft Excel and SPSS 20. For the collection and storage of data, a database was created and processed using the statistical program. Categorical data were described using frequencies and percentages. Descriptive statistics for continuous and discrete numeric variables and categorical variables are expressed as mean ± standard deviation and median (min–max).

### 2.9. Descriptive Analysis

We investigated three kinds of information: (i) patients (age, gender, personal and familial medico-surgical history and time between drug exposure and emergency department arrival), (ii) input services (medical discharge summaries and letters, and toxicological analyses) and (iii) events (categorized according to the World Health Organization adverse reaction terminology, WHO-ART).

Descriptive statistics are presented as the number and percentage for categorical variables and mean, standard deviation, minimum and maximum for numerical variables. Since the variables did not meet normal distribution conditions, independent two-group comparisons were performed using the Mann–Whitney U test. Chi-square analysis was used to evaluate categorical variable rates between the groups. Determining factors were examined using linear regression analysis.

The arithmetic mean (x) and standard deviation (SD) were calculated for the results of the two study groups at various time points. The level of statistical significance (α) was set as less than 0.05.

## 3. Results

### 3.1. Participants

In the present study, 250 patients were included, but after the data analysis, 43 of them were excluded because they also took other types of drugs, 35 also reported ethanol consumption and 8 patients were chronic opioid users. Figure 1 shows the STROBE flow diagram.

### 3.2. Descriptive Data

A total of 207 patients were enrolled in this study, 181 men (87.43%) and 26 women (12.57%) with an average age of 28.61 ± 9.31 (range 18–65 years), most being from the urban area (170, 82.12%), 35.74% of them being employed persons with a stable monthly income.

### 3.3. Main Results

The average time from consumption to admission to the emergency department was 2.8 h ± 2.47 h and 62.80% of the cases presented to the hospital by ambulance, being taken from their homes (51.69%). Most of the patients were admitted to the emergency room in the first hour after drug consumption (69 patients, 33.33%) and the predominant method of drug administration was smoking, as declared by 158 patients (76.32%). Among those who requested medical support, 108 patients (50.72%) declared occasional consumption, the triggering factor being curiosity. From the total of those enrolled in the study, 120 people declared that they started using out of curiosity, 49 of them declared that the people around them played an important role in the initiation of this vice and 22 patients declared the onset of drug use due to personal problems.

The Glasgow coma score (GCS) was calculated in all patients, evaluating the verbal, ocular and motor response spontaneously, to verbal stimulation and to the application of a painful stimulus. A score equal to or less than 8 points means a marked deterioration in the central nervous system and requires continuous monitoring of the patient with vital function support and airway assessment. In total, 5.79% (12 cases) of the patients included in the study had a GCS score ≤9 at ED admission, and eight of them required orotracheal intubation and mechanical ventilator support. All patients who required airway prosthesis were admitted to the emergency unit in the first three hours after consumption and had associated pathology.

Another clinical vital parameter that was evaluated is the respiratory function, recording the values of the respiratory rate (RR) and the oxygen saturation in the peripheral blood (SpO2). A total of 138 patients had respiratory rate values of over 20 breaths/minute, presenting tachypnea, corresponding to a percentage of 66.66%. The average value of the respiratory frequency was 21.40 ± 3.49, the minimum and maximum values being 14–26 breaths/min.

The oxygen saturation in the peripheral blood (SpO2) was obtained using the pulse oximetry technique. Following the statistical analysis, three peaks were recorded, one represented by a peripheral blood oxygen saturation value of 92%, a value found in 58 patients; a saturation value of 93% was recorded in 31 of the cases and a saturation value of 94% in 24 of those presented. The mean of the recorded values was 93.65% ± 2.55, the range being 90–100%.

The average value of the index perfusion tissue was 2.14 ± 2.16 (0.70–14).

All patients were evaluated in terms of the following hemodynamic parameters, monitoring blood pressure (BP) and heart rate (HR). The average values of the systolic blood pressure (SBP) were 125.23 ± 17.67 (85–196 mmHg) and of the heart rate were 87.48 ± 21.07.

### 3.4. Other Analyses

All the patients were divided into two groups: group A, which included all patients who presented in the emergency department within the first three hours after consumption, and group B, which included those patients who presented more than three hours and up to 12 h after drug consumption.

**Group A** included 163 people, the majority being male (146 men), most coming from the urban environment (135 cases). Most patients came to the emergency room in the first hour after consumption, complaining of dyspnea, psychomotor agitation, profuse sweating, palpitations and nausea. The oxygen saturation of the peripheral blood had an average value of 92.80 ± 1.86, the average values of SBP were 123.96 ± 17.50 and those of DBP were 74.85 ± 14.66. The average value of RR for group A was 22.5 ± 3.53 and in the case of heart rate, the average value was 87.83 ± 21.81. Upon admission to the emergency department, 158 patients from group A had a Glasgow coma score of over 8 points (mean GCS value 14.15 ± 2.52). The average values of the perfusion index in the case of group A were 1.51 ± 1.07 (0.70–5.50). Statistically significant correlations were found in the case of peripheral blood oxygen saturation (*p* < 0.001), respiratory rate (*p* < 0.001) and tissue perfusion index (*p* < 0.001).

**Group B** was represented by the group of patients who presented themselves or were admitted to the emergency department more than 3 h after drug consumption. The maximum time interval since consumption was 12 h (the average value being 6.44 ± 3.202). In this group, 44 people were included, the majority of them male (35 men and 9 women) and from the urban area (35 patients). At the time of the first medical consultation, those in group B presented average values of peripheral blood oxygen saturation of 96.86 ± 2.189 (*p* = 0.814) and average values of respiratory frequency of 17.12 ± 2.06. Regarding the neurological status, most patients had a GCS score >8 points at admission, the average values being 14.51 ± 1.47. From the cardiovascular point of view, the patients were generally hemodynamically stable, the average values of heart rate being 86.16 ± 18.13, SBP being 130.09 ± 17.65 and DBP being 78.86 ± 13.58. The average value of the tissue perfusion index was 4.55 ± 3.66 in the case of group B. From a statistical point of view, significant correlations in the case of this group were recorded between the respiratory rate at ED admission, the time between drug intake and emergency department presentation (*p* < 0.001) and tissue perfusion index (*p* < 0.001). In Table 1, the parameters recorded at the emergency department admission of the two groups of patients are represented.

## 4. Discussion

Cannabis use and its negative consequences have increased over the last several years in parallel with increasing cannabis potencies. SCs seem to be particularly popular among cannabis users [7].

Synthetic cannabinoids (SCs) emerged in the 1970s when researchers were first exploring the endocannabinoid system and attempting to develop new treatments for cancer pain. Around the year 2000, SCs appeared on the illicit drug market, where their prevalence had long been underestimated. Since then, their place in the market has steadily increased. Products of the same brand and sold under the same name have highly variable product compositions and concentrations [8,9].

This emerging market represents a specific public health problem in light of the severe complications in relation to their use. What the risks are of developing a psychotic disorder after SC administration remains a fundamental question [7].

The analysis of the specialized literature on this subject revealed that there are not enough previous studies to investigate the importance of the tissue perfusion index in patients presenting in emergency departments after drug exposure (synthetic cannabinoids, cannabis, marijuana and ethnobotanical substances). In a situation of low blood pressure and in the presence of circulatory failure, as a result of the vasomotor autoregulation that ensures the continuity of the blood flow, the direction of the blood flow changes from tissues of less vital importance to essential organs [10]. When the imbalance between the need and the supply of oxygen (O_2_) is prolonged, the vasomotor self-regulation mechanisms become insufficient and multiple organ failure syndrome sets in. The early determination of a low tissue perfusion index, in the absence of the activation of vasomotor self-regulation mechanisms, can be an early sign of multiorgan damage. Early identification of organ perfusion impairment plays an important role in preventing tissue hypoxia [11]. The average value of the perfusion index (PI) measured in patients in group A was significantly lower compared to that measured in patients in group B. Therefore, we concluded that the perfusion of peripheral organs and tissues was lower in the first 3 h after drug administration. PI values are obtained by measuring the ratio between the pulsatile signal during arterial flow and the non-pulsatile signal with a pulse oximeter, both values being obtained by means of absorbed infrared light [5].

Lima et al. compared capillary refill time, tissue perfusion index, and arterial oxygen saturation in patient and control groups and reported that hypoperfusion in critically ill patients was correlated with a PI value of 1.4 or less [12]. In another study by He et al. [13], a study on septic patients, the authors reported normal perfusion at a PI value of 1.4 or more, mild perfusion impairment between 0.6 and 1.4, and severe compromise perfusion at values of 0.6 or less [13]. PI values were slightly above the reported value of hypoperfusion in patients included in our study who presented to the ED within the first 3 h after using cannabis, marijuana, new psychoactive substances or synthetic cannabinoids. The average time from consumption to admission to the emergency department was 2.8 h ± 2.47 h. In total, 62.80% of cases presented by ambulance, 51.69% were taken from home and 33.33% presented themselves in the emergency room in the first hour after drug consumption.

However, the increase in the mean value of PI in our patients in association with the time elapsed until ED arrival raises the suspicion that the PI in the first minutes after drug use is lower than the determined mean value of PI and may also be below the critical value.

According to the literature, in some studies, it has been shown that the heart rate per minute in patients who use synthetic cannabinoids, cannabis, marijuana and ethnobotanicals can often remain within the normal physiological range after consumption, but cases of tachycardia, bradycardia or extrasystole arrhythmia have also been reported just as frequently [4,14,15]. In our study, heart rate per minute was within normal physiological limits both in patients who presented to the ED in the first 3 h and in those who presented after 3 h of consumption, and the difference between the two groups was statistically insignificant.

Another one of our findings was the fact that eight patients brought to the emergency department required orotracheal intubation and were admitted to the intensive care unit (ICU). also have been reported Hospitalization in the intensive care unit was not necessary for any patient in this study apart from these eight. In total, 5.79% of the patients included in the study had a GCS score ≤9 at the first medical evaluation and 66.66% of the patients had respiratory rate values of over 20 breaths/minute, presenting tachypnea.

No death was recorded, with all patients being discharged healthy. Most of the patients ended up in a psychiatric hospital, needing long-term counseling and support to combat drug addiction.

Monte et al. reported that most of the 76 users of synthetic cannabinoids (SC) included in their study were treated only in the ED, and only seven patients required admission to the ICU [16]. In another study by Küçük et al. [17], 54.1% of the patients presenting to the emergency room due to the consumption of cannabinoids required only symptomatic support, being discharged at home a few hours later after admission, while 24.1% of the patients required hospitalization in an intensive care unit. They attributed their low rate of home discharges to physicians’ low level of knowledge about the effects of cannabinoids [17].

We attribute the lower ICU admission rate and the lower mortality rate in the present study compared to previous research to the gradual familiarization of the emergency team with the spectrum of effects that can occur after cannabis-derived product and new psychoactive substance exposure.

In both groups in our study, the mean value of the Glasgow coma score was less than 15 points. In the case of group A, the average value was lower compared to group B. This may be due to the elimination in the following period, in the first 3 h after the use of cannabis and its derivatives, and, thus, to a weakening of their effects mediated by the CB1 receptor.

## 5. Conclusions

The tissue perfusion index shows the blood level of oxyhemoglobin and is an important parameter for the circulatory and respiratory system. PI plays an important role in the early detection of impaired organ perfusion and in monitoring tissue hypoxia. A decreased PI value may be an early indication of decreased perfusion organ damage.

The management of intoxicated patients has been described in numerous guidelines in order to standardize the emergency response, undergoing improvements over time compared to the results obtained so far. The initial approach is very important because it has an important contribution to decreasing morbidity and mortality, as well as to the long-term survival of patients.

The shortness of breath frequently reported by drug addict patients who request medical care after consumption, as well as the psychomotor agitation and the severity with which the symptoms are described when presenting to the emergency department, led to the decision to investigate the severity of these cases, using the index of tissue perfusion. There are very little data in the literature regarding the severity of cases of consumption correlated with the level of the drug in the body. The same can be said about the role of the tissue perfusion index in detecting severe complications in drug users. Based on this study, we can only say that there is a correlation between the tissue perfusion index, the oxygen saturation of the peripheral blood and the time since drug consumption. In our study, we cannot say that the value of the perfusion index can be correlated with the severity of the case, as the remission of the symptoms after a few hours and after supportive treatment was achieved in the majority of situations. Most of our patients were discharged home.

## Figures and Tables

**Figure 1 jpm-13-00372-f001:**
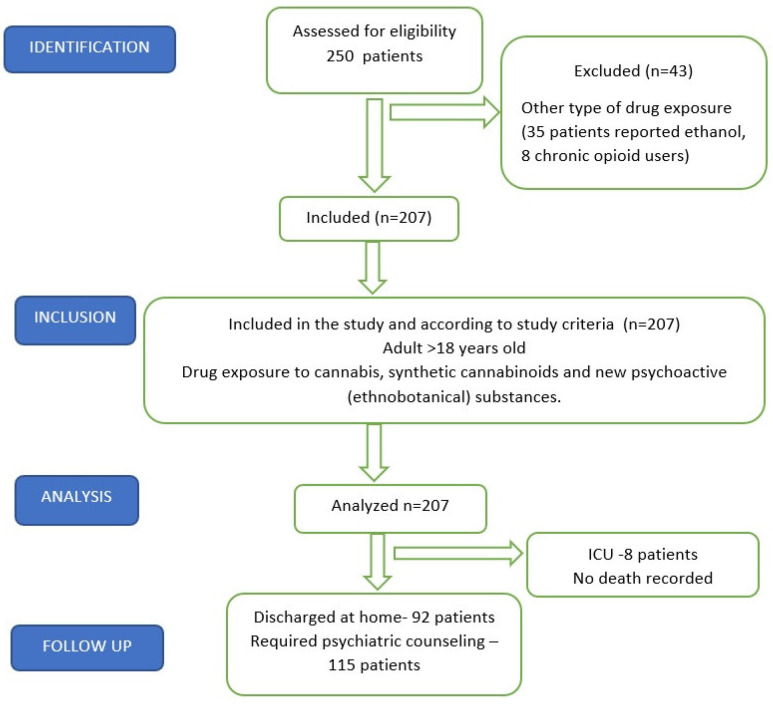
STROBE flow chart.

**Table 1 jpm-13-00372-t001:** Comparative analysis of the parameters recorded at the emergency admission of the two groups of patients.

	Gender M/F	Environment U/R	GCS Average	SpO2 Average	RR Average	HR Average	SBP Average	DBP Average	PI Average
Group A 1–3 h	146/17	135/28	14.15 ± 2.52	92.80 ± 1.86	22.52 ± 3.53	87.83 ± 21.81	123.96 ± 17.50	74.85 ± 14.66	1.51 ± 1.07
95% Confidence Interval of the Difference—group A			13.76–14.52	92.51–93.10	21.98–23.07	84.47–91.19	121.26–126.66	72.59–77.11	1.34–1.67
*p* value			*p* = 0.847	*p* < 0.001	*p* < 0.001	*p* = 0.492	*p* = 0.459	*p* = 0.419	*p* < 0.001
Group B 3–12 h	35/9	35/9	14.51 ± 1.47	96.86 ± 2.189	17.12 ± 2.06	86.16 ± 18.13	130.09 ± 17.65	78.86 ± 13.58	4.55 ± 3.66
95% Confidence Interval of the Difference —group B			14.06–14.96	96.19–97.53	16.48–17.75	80.58–97.14	124.66–135.53	74.70–83.02	3.50– 5.60
*p* value			*p* = 0.252	*p* = 0.814	*p* < 0.001	*p* = 0.246	*p* = 0.801	*p* = 0.583	*p* < 0.001

GCS—Glasgow coma score; SpO2—oxygen saturation of the peripheral blood; RR—respiratory rate; HR—heart rate; BPS—systolic blood pressure; DBP—diastolic blood pressure; PI—perfusion index, M—male; F—female; U—urban; R—rural.

## Data Availability

The raw data supporting the conclusions of this article are freely available from the authors upon request.

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
