# Peer review of "Monitoring Perfusion Index in Patients Presenting to the Emergency Department Due to Drug Use"

_jpm, 2023, doi:10.3390/jpm13020372_

Round 1
Reviewer 1 Report
This retrospective cohort study aimed to assess tissue perfusion in cannabinoid consumers. 207 patients were enrolled in the study, mostly men. The Authors found a significantly lower perfusion index within three hours of the drug intake. The study is interesting and provides clinical information about cannabinoid consumers.
However, I have two minor points:
I would suggest performing an analysis to assess if the PI can predict hospitalization in these patients.
If possible, I think that a figure could enrich the paper
Reviewer 2 Report
This manuscript does a good job demonstrating the important role in the early detection of impaired organ perfusion and in monitoring tissue hypoxia
Author Response
"Please see the attachment."

Reviewer 3 Report
Dear authors,
Thank you for this paper on an original subject. Despite its originality it does need major revision.
The structure of the article should be improved and contain clear sections and subsections.
Introduction: Overall too long with a lot of redundancy, few links between the physiology/physiopathology of the microcirculation issue and its relation to cannabis and SC intoxications. This part should highlight the reasoning of this postulated association, the gaps in the literature on this subject, the objective of the study. There are some missing references in citations.
Methods:
There are missing sub-sections. You could be inspired by using the STROBE guidelines :
- setting
- participants: eligibility criteria, sources and methods of selection
- variables and timing of their measurement
- data sources/measurements: electronic patient files, quality control, etc...
- bias
- study size : convenience sample? else?
- quantitative variables, grouping
- statistical analysis: to be improved, please better explain the descriptive analyzes according to the type of variable and the groups A and B, the comparisons of subgroups, and the correlation analyses (linear regression analyses, other??, this is unclear to me)
Results :
- too lenghty
- ad table 1 of "characteristics of patients" : it could be of benefit for better understanding of the cohort. Table 1 not clear (how were the p values calculated?). Mean GCS useful clinically (what about subcategories?)
- discussion : too lenghty and too many redundancies with the introduction, be careful not to overconclude, be careful not to ad new results not described before, no clear synthesis of the cited references
Round 2
Reviewer 3 Report
Dear authors,
I suggested some major revisions in my previous report. The new version does not demonstrate these, is it the right file?
I suggest also that you show clearly the modifications on the new upload in order to facilitate its review.
Best regards
